# Large-Scale Multi-Phase-Field Simulation of 2D Subgrain Growth

**Ali Khajezade [1], Warren J. Poole [1],\*, Michael Greenwood [2] and Matthias Militzer [1]**

[1] Department of Materials Engineering, The University of British Columbia, Vancouver, BC V6T 1Z4, Canada; xajezade@mail.ubc.ca (A.K.); matthias.militzer@ubc.ca (M.M.)

[2] Natural Resources Canada, CanmetMaterials, Hamilton, ON L8P 0A5, Canada; michael.greenwood@nrcan-rncan.gc.ca

\* Correspondence: warren.poole@ubc.ca

**Abstract:** The characteristics of subgrains in a deformed state after the high-temperature deformation of aluminum alloys control the subsequent recrystallization process and corresponding mechanical properties. In this study, systematic 2D phase-field simulations have been conducted to determine the role of deformed state parameters such as subgrain size and disorientation distributions on subgrain growth in an individual grain representing a single crystallographic orientation. The initial subgrain size and disorientation distributions have been varied by ±50%. To have a statistically relevant number of subgrains, large-scale simulations have been conducted using an in-house-developed phase-field code that takes advantage of distributed computing. The results of these simulations indicate that the growth of subgrains reaches a self-similar regime regardless of the initial subgrain structure. A narrower initial subgrain size distribution leads to faster growth rates, but it is the initial disorientation distribution that has a larger impact on the growth of subgrains. The results are discussed in terms of the evolution of the average diameter of subgrains and the average disorientation in the microstructure.

**Keywords:** grain growth; phase-field modeling; anisotropic subgrain growth; aluminum alloys; subgrain size distribution; disorientation distribution





## 1. Introduction

The microstructural characteristics of polycrystalline materials such as grain size distribution and crystallographic texture determine their mechanical and physical properties, such as flow stress, plastic anisotropy, and corrosion behavior. This study is concerned with microstructure evolution in high stacking fault metals and alloys, specifically aluminum alloys after large strain deformation at high homologous temperatures, i.e., 60–85% of the melting point. These conditions are particularly relevant to the high-temperature extrusion of aluminum alloys, which are increasingly being used in automotive applications to reduce vehicle weight.

The evolution of microstructures at high temperatures depends critically on the characteristics of the deformed state, as the mechanical work performed on the material is partially stored in the deformed structure, which can be characterized by multiple factors, including grain shape, grain orientation, and the distribution of the dislocation substructure within the grains. There is a complex interplay between the deformation path, temperature, strain rate, and the development of the local microstructure, which leads to different final microstructural characteristics.

While the grain boundary area and the evolution of grain orientations can be reasonably predicted as a function of the applied strain and the local strain path using continuum-based polycrystal plasticity models [1–3], the prediction of the details of dislocation substructure within the grains is more complicated (only possible in specific cases under significant assumptions, e.g., discrete dislocation dynamics simulations at low strain [4,5]). Under

suitable conditions, however, recrystallization can be suppressed during high-temperature deformation to preserve the deformed state for subsequent experimental characterization. This bypasses the problem of predicting the deformed state, allowing for a focus on the evolution of the subgrain distribution during annealing which, for example, in a number of aluminum alloys, leads to the recrystallized microstructure. It has been found that the dislocation substructure may depend on the grain orientation [6–10]. As reported in these studies, the average size and disorientation of subgrains are different within each texture component and grain orientation, and they depend on the deformation conditions. As a result, there can be significant local variations in the stored energy from grain to grain in the deformed state. For materials such as aluminum alloys deformed to large strains above 350 °C, the dislocation substructure relaxes into cell substructures with a low density of dislocations in the interior of the cells and low-angle boundaries between subgrains formed by dynamic recovery during the deformation [11,12]. The kernel average misorientation (KAM) analysis from high-resolution Electron Backscatter Diffraction (EBSD) measurement conducted by Chen [10] on the deformed state of an Al–Mn–Fe–Si alloy extruded at high temperature clearly shows that the material is dynamically recovered into a very well-defined cell structure (i.e., subgrains) with little density of free dislocations within the cells, and the characteristics of the dislocation cell structures are different within different grain orientations. While there are some free dislocations within the subgrains, their contribution to the stored energy is negligible compared with the subgrain boundaries. Thus, the stored energy in the material can be approximated by the interfacial energy of subgrains and grain boundaries, and the contribution of stored energy within the subgrains can be ignored. The characteristics of these substructures, such as subgrain size and disorientation distribution, depend on the crystallographic orientation of the initial grain and how its orientation evolves during deformation [10,13,14].

The current study emphasizes the microstructure evolution after axisymmetric extrusion of an Al–Mn–Fe–Si alloy at high temperatures (350 °C) to a logarithmic strain of ~4. Chen has previously reported that by using an appropriate homogenization heat treatment prior to extrusion, a high density of Mn-based dispersoids can be formed, which suppresses recrystallization via Smith–Zener pinning and preserves the deformed state after high-temperature extrusion [10]. Chen reported highly elongated dynamically recovered grains, with more than 98% of the grains having a specific orientation relationship with the extrusion direction, ED, i.e., either <001>||ED or <111>||ED orientations, consistent with other reports for axisymmetric deformation of FCC metals [15,16]. In a preliminary analysis, Chen found that grains oriented with <001>||ED had larger average subgrain sizes, as well as larger average disorientations as compared with those with <111>||ED [10].

The energy stored in the deformed state provides the driving force for the restoration phenomena, such as recovery and recrystallization. The restoration phenomena can be phenomenologically categorized into continuous and discontinuous based on the ability to distinguish the nucleation stage from the growth stage [17]. Recrystallization in aluminum alloys deformed at high temperatures (such as extrusion) takes place as an extended recovery process, which involves subgrain coarsening to reduce the stored energy in the material [18–20].

The subject of grain growth has been studied using both analytical (e.g., [21–24]) and full-field numerical approaches (e.g., [25–28]). Generally, the analytical approaches average the microstructure based on the statistical information from microstructural features, while the full-field approaches explicitly describe the microstructure based on its building blocks, such as grains and their interfaces, which allows for more sophisticated assumptions and more accurate predictions. While all the numerical techniques have their own benefits and drawbacks, phase-field modeling is used in the current research as the physical mechanisms can be explicitly included in the framework without the need to track the interfaces.

It is well established in the literature on grain growth that the incorporation of grain boundary characteristics such as inclination angle and disorientation can have significant impacts on the evolution of the grain structure and how local microstructures evolve within

a deformed grain (e.g., [29–31]). Some studies [32,33] have assumed two types of boundaries, i.e., boundaries with high effective mobility and boundaries with low effective mobility. On the other hand, there are studies [28,34–37] where a distribution of boundary energy and mobility was assumed. In the majority of these studies, Huang–Humphreys [38,39] and Read–Shockley [40] models were used for the mobility and interfacial energy of low-angle boundaries, respectively. In a few other studies, phase-field modeling was coupled with databases generated from molecular dynamics simulations for the mobility and interfacial energy of the boundaries [34,35]. Despite the significance of boundary characteristics, most of these studies assume disorientation distributions that are not relevant to the characteristics of subgrains in the deformed state (e.g., very weak or random texture), and only a few studies are focused on this issue [37,41–45].

In this regard, Holm et al. [37] simulated microstructure evolution with a narrow disorientation distribution and observed a continuous increase in the probability density of disorientations of fewer than 2° during anisotropic subgrain growth, and the growth regime never reached a steady state. Gruber et al. [41], Esley et al. [42], Zöllner et al. [43], and Niño et al. [44,45] studied the role of disorientation distribution on subgrain growth of textured materials and observed that the disorientation distribution changes favored low-angle boundaries. It is worth noting that these simulations were conducted on relatively small domains, and there is a need to address the statistical representativity of the simulated domains. Thus, more realistic disorientation distributions and large statistically relevant simulation domains are required.

In the current study, the role of initial subgrain size distribution and initial disorientation distribution have been systematically evaluated using synthetic microstructures with statistically relevant domain sizes, i.e., from ~400,000 to 1,100,000 initial subgrains. The baselines for initial distributions for subgrain size and disorientations were taken from the dataset provided by Chen [10]. The results of 2D phase-field simulations for systematically varied initial subgrain structures are discussed to evaluate the microstructure evolution in terms of deformed state parameters, i.e., initial subgrain size and disorientation distributions. The long-term goal is to establish a framework for the interaction between industrial processing conditions (local deformation path, temperature, and strain rate within the extrusion) and alloy chemistry and pre-extrusion thermal history (solute and second-phase particles and thermal history) on the evolution of microstructures after extrusion. This paper addresses how the details of the local deformed state affect subgrain growth in a single deformed grain.

## 2. Methodology

### 2.1. Microstructure of Deformed State

The microstructure of the deformed state and the recrystallized state were reanalyzed from the dataset for extruded samples provided by Chen using EBSD studies [10]. The EBSD data were cleaned and analyzed using the MTEX toolbox (Version 5.6.0) [46] in MATLAB(Version R2022b, MathWorks, MA, USA). All data points with a low confidence index (CI < 0.15) were removed from the dataset (82% of the data points were left). Then, grains were identified using a threshold disorientation angle of 15°. Using this analysis, small grains with an area of fewer than 20 pixels were removed. Any area with a distinct orientation within each individual analyzed grain was assumed to be a subgrain. After this step, a half quadratic filter [47] with a smoothing factor of 0.1 was used to reduce noise and to populate the nonindexed pixels. Finally, a hexagonal grid was mapped to a rectilinear grid with 200 nm resolution.

Examples of the elongated grains were identified, and then grains with <001>||ED and <111>||ED were isolated to characterize the subgrain size distribution and disorientation distribution. The subgrains were identified with a threshold angle of 2°, i.e., each pixel in the EBSD map was calculated relative to its neighbors, and if a pixel had a disorientation larger than 2°, it was assigned to its associated subgrain boundary. All the histograms of disorientation distributions in this study are the pixel-weighted disorientation distribution.

As the grains were identified using a threshold disorientation angle of 15°, the subgrains can have orientations that are rotated up to ±15° with respect to the mean orientation of the grain. Consequently, the disorientation distribution of subgrains can have disorientations of up to 30°. A lognormal distribution was fit to the subgrain size distribution for each of the individual isolated grains. These subgrain size and disorientation distributions served as the baseline cases. The mean and standard deviation from these fittings were used as input to the Neper software (Version 4.1.3-10) to create synthetic microstructures. An orientation distribution function (ODF) was fit to the orientation measurements for the subgrains in the two elongated grains using the direct kernel density estimation method [48]. Cubic crystal symmetry was enforced in the ODF calculations, and no sample symmetry was enforced.

## 2.2. Phase-Field Modeling and Implementation

To simulate the evolution of the grain structure, a phase-field model based on the formulation proposed by Steinbach et al. [25,49] was built. In this model, the subgrains were described by continuous field variables, $\phi_i$, which are defined over the entire simulation domain ($\phi_i$ is called the order parameter of the $i$th grain). The index $i$ refers to each subgrain, and it starts from 1 to the maximum number of subgrains inside the simulation box ($N_{max}$). The sum of all order parameters at a specific position must be 1, i.e., $\sum_{i=1}^{N_{max}} \phi_i = 1$. Using the order parameters and their gradients, the state of the system can be described by a free energy functional as follows:

$$\mathcal{F}(\{\phi_i\}, \{\nabla\phi_i\}) = \int \sum_{i,j}^{v} \frac{4\sigma_{ij}}{\eta_{ij}} \left( -\frac{\eta_{ij}^2}{\pi^2} \nabla\phi_i \cdot \nabla\phi_j + \phi_i\phi_j \right) dV \tag{1}$$

where the brace bracket, {}, represents all constituents contributing to the total free energy functional. Note that other forms of energy (e.g., chemical energy) could be included in the framework, but since there is no spatial variation in these other energy terms, they can be ignored in the present study. Here, $v$ is the local number of order parameters, i.e., subgrains at a given position in the system, $\sigma_{ij}$ is the interfacial energy between subgrains $i$ and $j$, and $\eta_{ij}$ is the interface width. The governing multi-phase-field equation for the evolution of the system can be derived as

$$\dot{\phi}_i = \sum_{j=1}^{v} \frac{M_{ij}}{v} \left( \sum_{k=1}^{v} \left( \sigma_{jk} - \sigma_{ik} \right) \left( \nabla^2\phi_k + \frac{\pi^2}{\eta^2}\phi_k \right) \right) \tag{2}$$

where $M_{ij}$ is the mobility of the interface between subgrains $i$ and $j$.

In the current study, the Read–Shockley equation [40] is used for the interface energy, $\sigma_{ij}$, between subgrains $i$ and $j$, i.e.,

$$\sigma_{ij} = \sigma_{\theta_{th}} \frac{\theta_{ij}}{\theta_{th}} \left( 1 - \ln(\frac{\theta_{ij}}{\theta_{th}}) \right) \tag{3}$$

where $\theta_{ij}$ is the disorientation between subgrains $i$ and $j$, and $\theta_{th}$ is the threshold angle defined for the transition to a high-angle boundary. $\sigma_{\theta_{th}}$ is the interfacial energy for high-angle boundaries. The threshold angle was defined as $\theta_{th} = 15°$, and a value of $\sigma_{\theta_{th}} = 3.24 \times 10^{-1} \frac{J}{m^2}$ was assumed [50]. A modified version of the Huang–Humphreys equation was used for the interface mobility [38,51]. Here, it was assumed that the boundaries with disorientations of fewer than 2° have the same mobility as 2° boundaries:

$$M_{ij} = \begin{cases} M_{\theta_{th}} \left\{ 1 - exp\left[ -5\left(\frac{\theta_{ij}}{\theta_{th}}\right)^4 \right] \right\}, & \theta \geq 2° \\ M_{2°}, & \theta < 2° \end{cases} \tag{4}$$

where $M_{\theta_{th}}$ is the mobility of high-angle boundaries and $M_{2^\circ}$ was the mobility of boundaries with a disorientation angle of $2^\circ$. A value of $M_{\theta_{th}} = 5 \times 10^{-11} \, \frac{m^4}{Js}$ was adopted for all simulations. This value was approximated from the data for an Al–0.05 wt% Si alloy at 400 °C, as provided by Huang and Humphreys [39]. Note that in the current investigation, an effective mobility was assumed (refer to [52] for appropriateness of the assumption); however, solutes such as Mn in Al–Mn–Fe–Si systems may have an influence on the mobility of the interfaces. For example, Mn can reduce the migration rates of subgrain boundaries by solute drag and the formation of dispersoids (Mn precipitates). The implementation of a cut-off angle for the mobility, i.e., $2^\circ$, was employed to reduce the numerical inaccuracy in situations where a boundary with extremely low mobility meets with other boundaries at a junction [53].

The model was coded in the C++ programming language using a Message Passing Interface (MPI) approach based on the architecture developed by Greenwood et al. [54]. Figure 1 shows a high-level representation of the code structure.

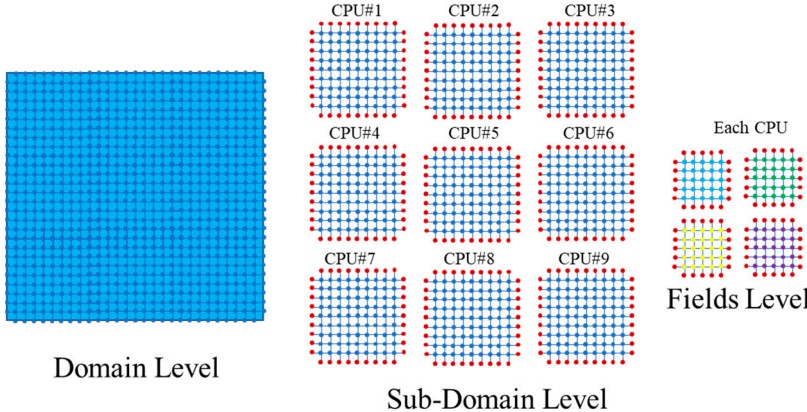

**Figure 1.** A schematic representation of the approach taken in the current work for the message parsing interface (MPI) communication. Red points are buffer nodes for MPI communications, other colors are used for simulation nodes. Different colors are used for different field levels.

As shown, the code splits the simulation domain into smaller subdomains, which are distributed over several CPUs on a cluster of computational resources. Each subdomain is divided further into fields. Fields have an extra set of nodal points around them, which are called buffer nodes. These nodes are utilized to inform field boundary nodes (nodes located on the edges of the field) about information regarding the neighboring nodes in surrounding fields (which can be present in a subdomain assigned to another CPU). In the first step, the code communicates between fields and subdomains to fill these buffer nodes, then the multi-phase-field model is solved in each of these fields independently. This process can be repeated for any desired number of iterations to achieve the intended total simulation time.

The code tracks possible subgrains and their number at each grid point, i.e., $\nu$ in Equation (2), and solves this equation only for order parameters associated with these subgrains to avoid solving for millions of grains at each grid point. The code architecture also allows for the local definition of subgrain interactions at the field level for the definition of interface disorientations for interfacial mobility and energy calculations. This is necessary because defining pair-wise interactions globally for overall microstructure would be computationally prohibitive due to the large number of subgrains involved. By defining a table of interaction for each field, which contains significantly fewer subgrains, the tables can be dynamically updated after each step of the solver for adding or subtracting new subgrain interactions by checking the buffer nodes.

The finite difference method was used to solve the system of equations, i.e., Equation (2). The Laplacian term was approximated by a five-point stencil formulation. Sensitivity analyses were conducted regarding the time increment and grid spacing to ensure the accuracy of multi-phase-field simulations (see ref. [55] for details). After the sensitivity analysis, the mesh resolution was chosen to be 0.25 μm (with 97% accuracy), and the interface width was taken to be 6 cells, i.e., 1.5 μm. For time integration, a forward Eulerian scheme was used with a time increment of $5 \times 10^{-5}$ s. The total simulation time was selected to be 7.5 s for all simulations to study the long-term behavior of (sub)grain growth, i.e., final grain sizes of 30–40 μm, which are typical of recrystallized aluminum alloys.

### 2.3. Generation of Synthetic Microstructures

The subgrain structure of synthetic microstructures for baseline conditions was generated with the Neper software package [56]. The mean and standard deviation measured from the lognormal fit to the experiments provide the inputs for the Neper software. The orientations for each of the synthesized baseline microstructures were generated using MTEX toolbox [46] in MATLAB. For this purpose, an ODF was fit to the experimentally measured orientations within the two isolated grains based on the direct kernel density estimation method with a specified halfwidth angle equal to the average disorientation calculated from the experimental measurement. Then, the orientations were randomly generated from the calculated ODF.

To systematically study the role of the initial subgrain structure on the evolution of subgrain growth, the Neper software was used to generate different initial microstructures by altering the widths of the standard deviation for subgrain size and disorientation distributions. In detail, the subgrain size distribution was altered by ±50% for the baseline condition while otherwise maintaining the baseline disorientation distribution. Further, the initial disorientation distribution was altered using the baseline conditions for subgrain size distribution and changing the halfwidth angle of the baseline ODF by ±50% in 25% increments. In this study, any area in the synthetic microstructure with a distinct orientation is defined as a subgrain regardless of its surrounding boundary disorientations. It is worth mentioning that the classical definition of a grain is any area surrounded by high-angle boundaries (disorientation > 15°); thus, there is a small probability that the synthesized subgrains fall under this definition. However, their probability is very low, as the chosen halfwidth angles for the generation of disorientation distributions are less than 15°.

The size of the simulation domain was chosen to ensure a statistically representative number of subgrains at the end of the simulation. Approximately 10,000 distinct grain orientations need to be sampled in order to achieve a statistically relevant distribution to quantify the crystallographic texture of the material [57,58]. Given the experimentally measured average recrystallized grain diameter of 14.6 μm [10], a representative domain size of 2048 μm × 2048 μm was determined. This led to an initial number of subgrains of ~400,000 to 1,100,000, depending on the details of the initial subgrain size distribution.

To generate synthetic microstructures in a computationally acceptable time for such a large domain size, a multi-scale tessellation approach was used [59]. The 2D domain was split into 256 square subdomains, and the distribution of subgrains was generated within each subdomain. The coordinates of cell seeds and their associated weights for the tessellation in the previous step were input into Neper to generate a single large microstructure, as summarized in Figure 2. Ten initial increments of phase-field calculations with a smaller time increment (10 times smaller than the stable maximum time increment for the explicit finite difference technique) were used to let the boundaries relax toward their equilibrium state at the beginning of the simulations.

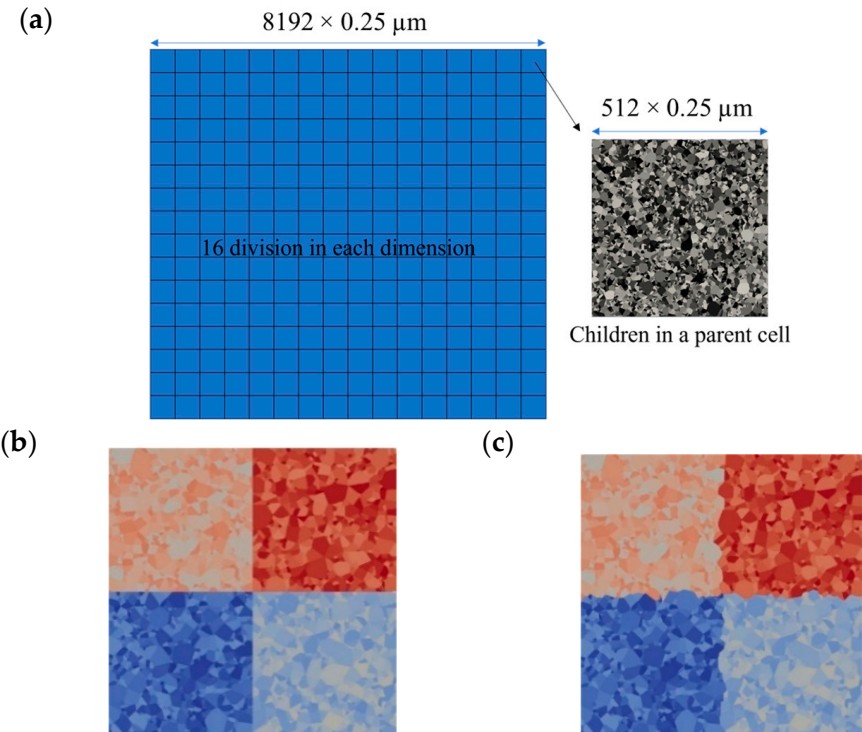

**Figure 2.** (**a**) Subdivision of a large 2D domain into 256 divisions for a faster microstructure generation (256 times faster than undivided domain); (**b**) an example of microstructure that is generated after doing tessellation in a divided domain; (**c**) final step where the seeds and their weights are input to Neper to anneal out the straight boundaries between parent cells. Note: the colors in (**b**,**c**) represent subgrain IDs within each distinct subdomain, and they were used to improve the visualization of boundaries between subdomains before and after annealing.

## 3. Results

### 3.1. Initial Microstructures

3.1.1. Baseline Conditions

Figure 3a shows an experimentally measured EBSD inverse pole figure (IPF) map of the extruded sample. The two isolated elongated grains are shown in Figure 3b,c, i.e., one with a predominate grain orientation of <001>||ED (red) and one with <111>||ED (blue), respectively. The subgrain disorientation and size distributions are shown in Figure 4a,b, respectively. The average disorientation is 9° for subgrains oriented with <001>||ED and 6.9° for subgrains oriented with <111>||ED.

Figure 4b shows the fit of a lognormal distribution to the subgrain size distribution:

$$f(x) = \frac{1}{x\sqrt{2\pi\ln\left(1+\frac{\sigma^2}{\mu^2}\right)}}\exp\left\{-\frac{\left(\ln(x)-\ln\left(\frac{\mu^2}{\sqrt{\mu^2+\sigma^2}}\right)\right)^2}{2\ln\left(1+\frac{\sigma^2}{\mu^2}\right)}\right\} \tag{5}$$

where $x$ is the variable, $f$ is the probability density function, and $\mu$ and $\sigma$ are the average and the standard deviation, STD. The average subgrain diameter was found to be 2.06 μm and 2.66 μm with a standard deviation of 1.42 μm and 1.65 μm for the <111> ||ED and <001> ||ED grains, respectively. These values were used as baseline conditions for the simulations in this study.

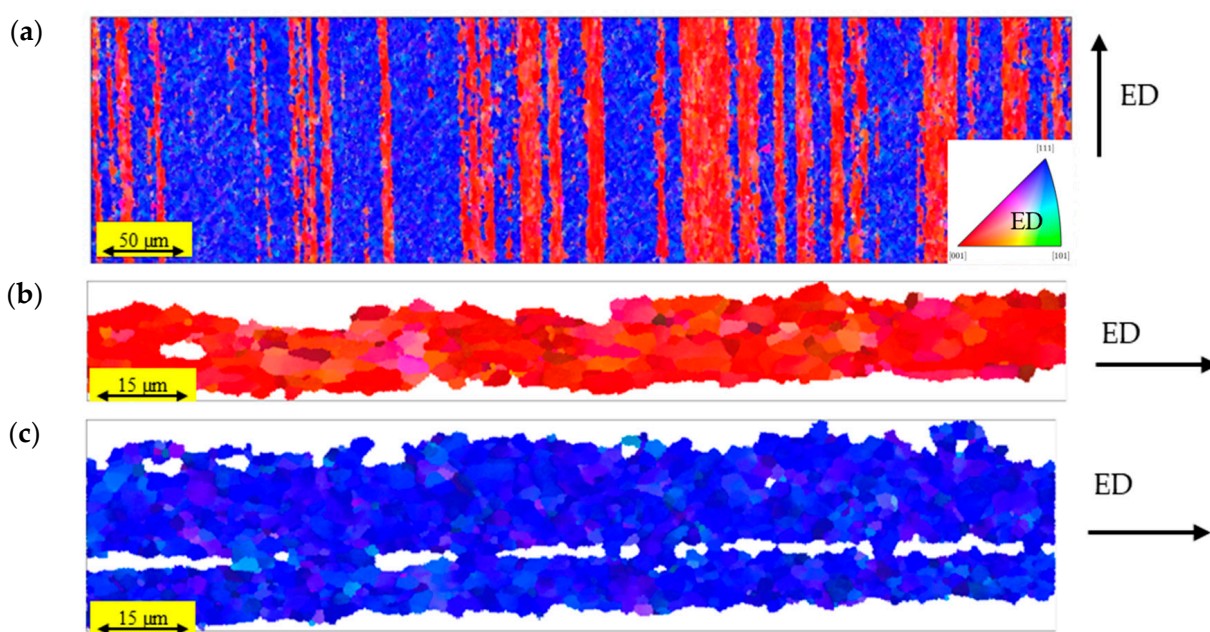

**Figure 3.** (**a**) an EBSD IPF map of the grain structure for the Al–Mn–Si–Fe alloy homogenized at 375 °C for 24h and extruded at 350 °C; (**b**) a grain with <001>||ED orientation; (**c**) a grain with <111>||ED orientation.

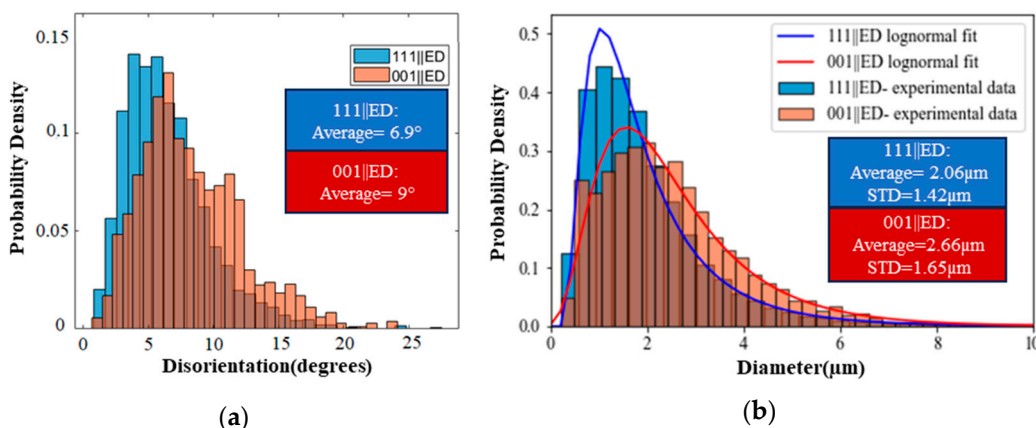

**Figure 4.** (**a**) The disorientation distribution within <001>||ED and <111>||ED; (**b**) subgrain size distribution within grains with <001>||ED and <111>||ED.

### 3.1.2. Result of Phase-Field Simulation for the Baseline Conditions

Figure 5 shows the final subgrain structures obtained from phase-field simulations for baseline conditions. As shown in Figure 5a,b, the microstructures for the <111>||ED and <001>||ED subgrains are similar. The average 2D subgrain diameters for <111>||ED and <001>||ED are ~33 μm and ~36 μm, respectively, and the evolution of the average diameter tends to follow a similar trend for the two different types of grains. Minor differences can be attributed to the relatively small difference in the initial subgrain size and disorientation distributions.

Given the similarity of the results, only the simulation results of <111>||ED are shown in the following. The interested reader can find the results for <001>||ED in the Supplementary Materials. It should be noted, however, that when grains of both orientations coexist, this difference can be important.

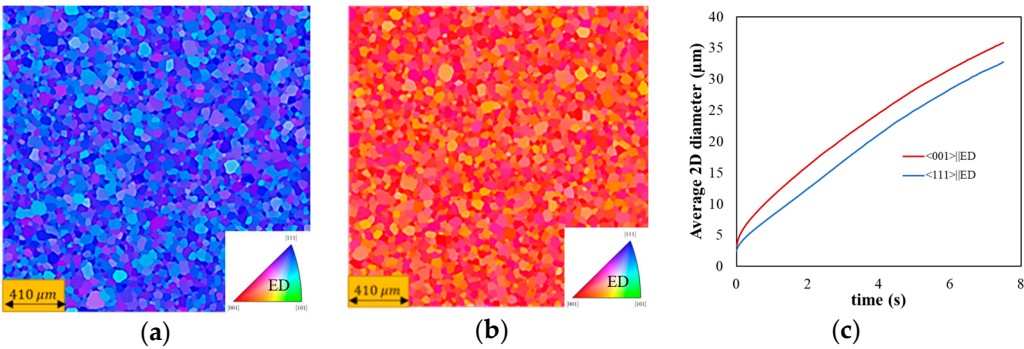

**Figure 5.** The microstructures at the end of the simulation for (**a**) <111>||ED and (**b**) <001>||ED, and (**c**) the average grain size vs. simulation time for the two different initial microstructures.

3.1.3. Modification from Baseline Conditions

The first set of synthetic microstructures was generated to systematically determine the role of the initial subgrain size distribution on the subsequent microstructure evolution. Figure 6a shows the different initial subgrain size distributions for <111>||ED, and Table 1 summarizes the key parameters for each initial microstructure.

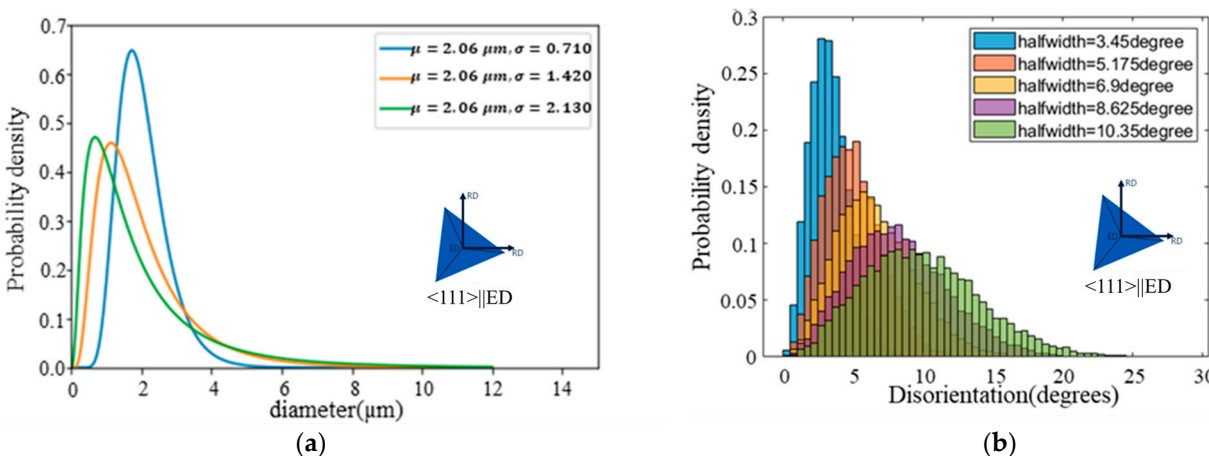

**Figure 6.** Different simulations to study the role of (**a**) initial subgrain size distribution and (**b**) disorientation distribution on growth. See Supplementary Figures S1 and S2 for the initial conditions for <001>||ED grain and initial generated microstructures.

**Table 1.** Key parameters for the initial microstructures synthesized to study the role of subgrain size distribution (see Table S1 in the Supplementary Materials for <001>||ED).

| Orientation | 50% Narrower | Baseline | 50% Wider |
|---|---|---|---|
| <111>||ED | $\mu = 2.06$ μm, $\sigma = 0.71$ μm | $\mu = 2.06$ μm, $\sigma = 1.42$ μm | $\mu = 2.06$ μm, $\sigma = 2.13$ μm |

A second set of microstructures was generated to study the role of subgrain disorientation distribution while keeping the subgrain size distribution constant. Figure 6b shows different initial disorientation distributions for <111>||ED, and Table 2 summarizes the key parameters for each initial microstructure.

**Table 2.** Half-width angle in degrees for ODF calculation of the initial microstructures synthesized to study the role of disorientation distribution (See Table S2 in Supplementary Materials for <001>||ED).

| Orientation | 50% Narrower | 25% Narrower | Baseline | 25% Wider | 50% Wider |
|---|---|---|---|---|---|
| <111>||ED | 3.45 | 5.17 | 6.9 | 8.62 | 10.35 |

### *3.2. Role of Initial Subgrain Size Distribution*

Figure 7 shows the simulated evolved microstructures using the different initial subgrain size distributions after a simulated time of 7.5 s. While qualitatively similar, the evolution of the average equivalent area diameter and the growth rates show differences (see Figure 8a,b, respectively). For the 50% wider distribution, the subgrain size initially increases at a higher rate before the rate slows down significantly such that the final microstructure has an average diameter that is ~18% smaller than the baseline case. On the other hand, when the initial subgrain size distribution is 50% narrower, the average subgrain growth rate is slower at short times (<0.5 s), but the final subgrain diameter is ~8% larger than the baseline case. The increased growth rate for the 50% narrower microstructure occurs from 0.05 s to about 1 s. Despite these differences in the initial growth rates, all three cases approach a growth rate plateau with values between 2–4 μm/s, as seen in Figure 8b. Figure 9 shows the normalized subgrain size distributions after 5, 6.25, and 7.5 s of simulation time, where it can be observed that self-similar growth is occurring, i.e., there is no change in the normalized subgrain size distribution.

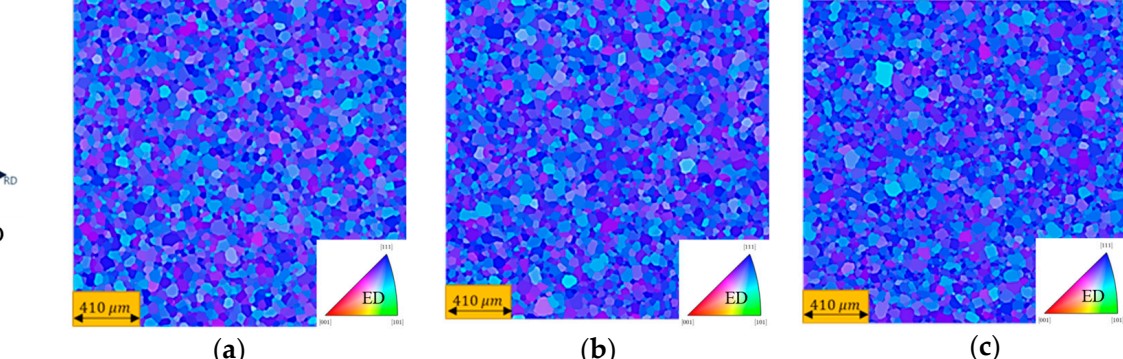

(**a**)   (**b**)   (**c**)

**Figure 7.** The subgrain structure at the end of the simulation for different initial microstructures, i.e., subgrain size distributions: (**a**) 50% narrower, (**b**) baseline, and (**c**) 50% wider (note: Figure S3 in the Supplementary Materials shows the results for <001>||ED).

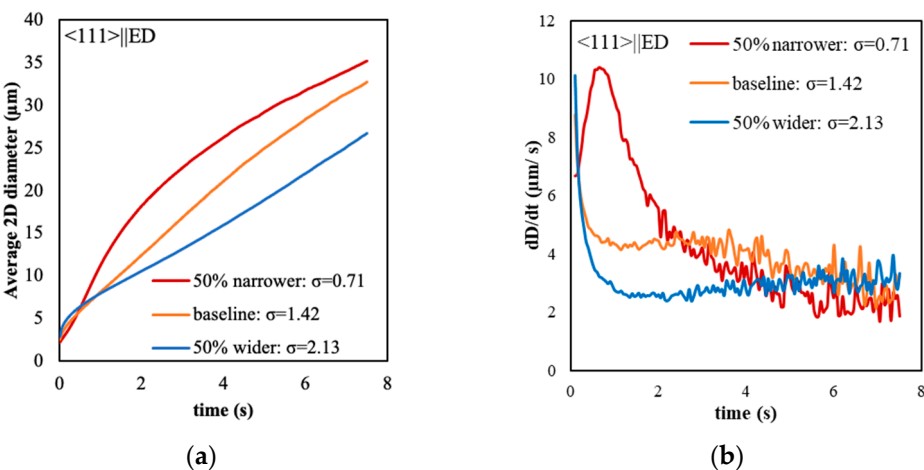

(**a**)       (**b**)

**Figure 8.** (**a**) Evolution of the equivalent area average 2D diameter of <111>||ED microstructures with different initial subgrain size distribution; (**b**) evolution of the rate of change in the equivalent area average diameter of the same microstructures (note: Figure S4 in the Supplementary Materials shows the results for <001>||ED).

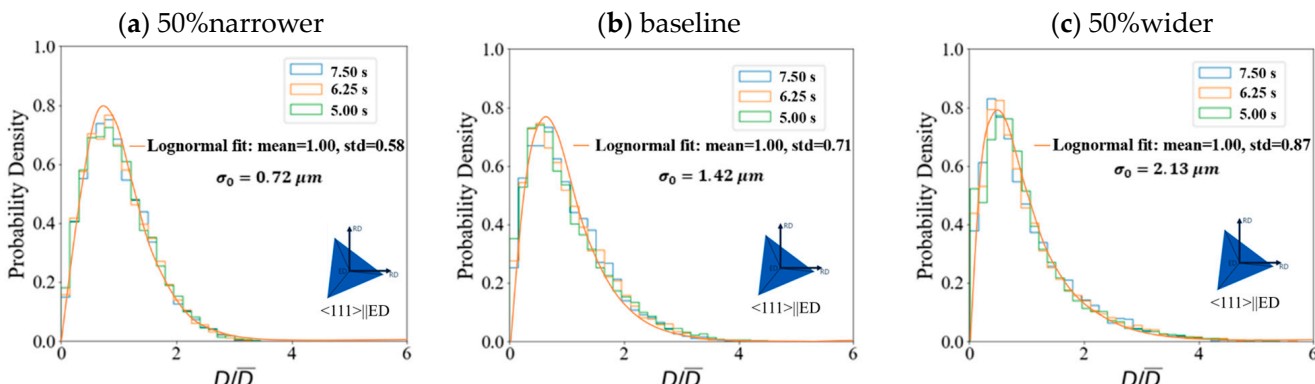

**Figure 9.** Histogram of evolved normalized diameter for microstructures with different initial standard deviation and same initial average diameter in different time steps: (**a**) 50% narrower, (**b**) baseline, and (**c**) 50% wider standard deviation (note: Figure S5 in the Supplementary Materials shows the results for <001> | | ED).

In detail, the scaling distributions are different. As the initial subgrain size distribution gets wider, the distribution of the normalized diameter gets wider, and the peak of the normalized diameter distribution shifts to higher values. The standard deviation of the normalized diameter for the 50% narrower size distribution, the baseline, and the 50% wider size distribution is 0.58, 0.71, and 0.87. The distribution of the normalized diameter is positively skewed with a comparatively sharp peak, and the position of the peak is at $\frac{D}{\bar{D}} < 1$. These distributions can be contrasted to that of ideal normal grain growth where 2D simulations result in a symmetric distribution around $\frac{D}{\bar{D}} = 1$ with a flat-shaped peak (e.g., [35,60]). The asymmetry of the subgrain size distribution in the current study has also been found in other studies (e.g., [61]).

To rationalize these differences, the evolution of the total length, as well as the fraction of high-angle boundaries (i.e., disorientation $\geq 15°$), may be considered, as illustrated in Figure 10a,b. The total length of the high-angle boundaries initially shows (except for the 50% tighter subgrain size distribution) a small decrease which, as suggested in reference [37], is related to shrinkage of small subgrains surrounded by boundaries with higher-than-average effective mobility (i.e., interface mobility times interface energy). As can be seen from Figure 10, there is a small drop in the total length of the high-angle boundaries (which leads to a drop in the average disorientation) at the beginning of the simulation for the base and wider subgrain size distributions, suggesting that there is a higher probability of subgrain shrinkage and disappearance. By careful examination of the initial subgrain size distribution in Figure 6, it can be observed that the chance of finding very small subgrains ($d < 0.5$ μm) is negligible for narrow distributions, while this is not the case for wider distributions. Setting aside the initial decrease, one can observe that the rate of increase in the high-angle boundary fraction (and in the associated average disorientation) increases as the width of the subgrain size distribution decreases. For the narrow distribution, the total length of the high-angle boundaries initially increases, but after reaching a maximum, it decreases. For the other subgrain size distributions, the length increases after the initial transient but apparently does not reach a maximum within the investigated simulation times.

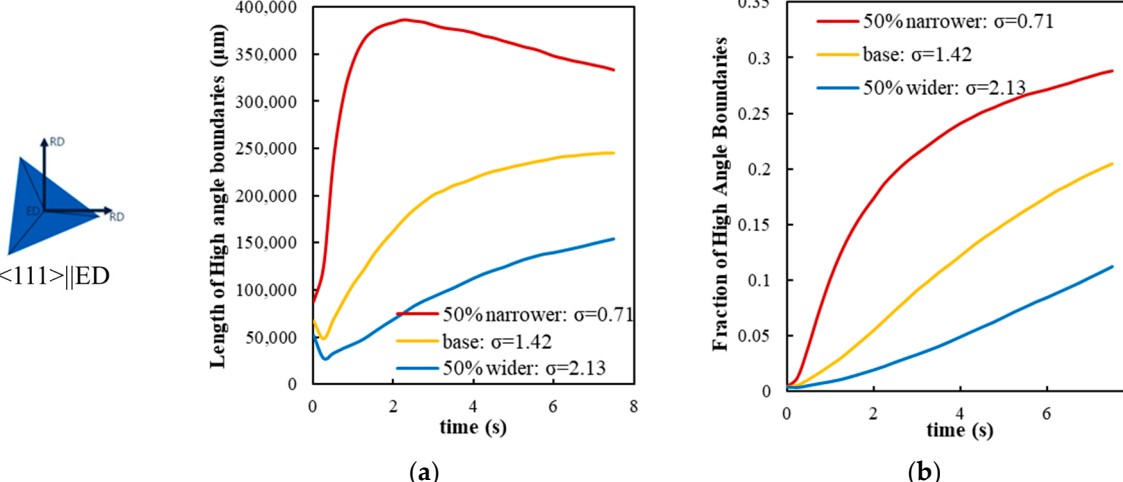

**Figure 10.** Evolution of (**a**) the total length of high-angle boundaries and (**b**) the fraction length of the high-angle boundaries of subgrains <111>||ED (note: Figure S6 in Supplementary Materials shows the results for the same analysis in the case of <001>||ED).

### 3.3. Role of Initial Disorientation Distribution

Figure 11 shows the microstructures for simulations after 7.5 s for different initial disorientation distributions for the <111>||ED grain, and Figure 12 illustrates the evolution of the average disorientation. As the width of the initial disorientation distribution increases, the subgrain structure coarsens at an increasing rate, resulting in a higher fraction of high-angle boundaries (increased average disorientation in Figure 12). The biggest difference in the growth trends can be observed when the half-width angles are lower than those of the baseline, as shown in Figure 11d. For half-width angles larger than 10°, the trends of subgrain growth approach an upper bound. After an initial short-term drop in the average disorientation distribution, it increases with a rate that increases with the width of the disorientation distribution. As mentioned previously, the initial drops are proposed to be related to the shrinkage of small subgrains surrounded by boundaries with higher-than-average effective mobility, as seen in other studies [37]. It is noted that the fraction of high-angle boundaries is lower than expected from thermodynamic considerations, but one expects that given sufficient simulation time, this fraction would increase.

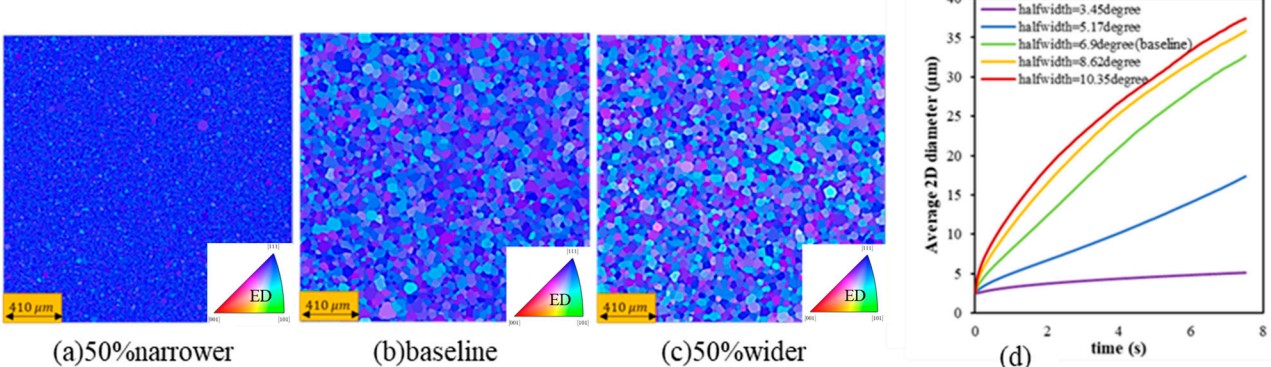

**Figure 11.** The evolved microstructure of a grain in <111>||ED fiber with the same subgrain size as baseline and disorientation distribution of different half-width angles (**a**) 3.45°, (**b**) 6.9° (baseline), and (**c**) 8.62° at 7.5 s and (**d**) evolution of diameters for different half-width angles (note: Figure S7 in Supplementary Materials shows the results for <001>||ED).

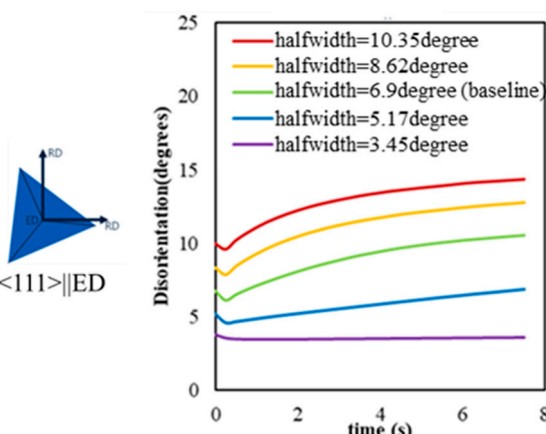

**Figure 12.** The evolution of average disorientation in the different microstructures of different disorientation distributions with the same subgrain size in a grain with <111>||ED (note: Figure S8 in Supplementary Materials shows the results for <001>||ED).

## 4. Discussion

### 4.1. Role of Initial Subgrain Size Distribution

The simulation results for the subgrain size evolution and the rate of growth shown in Figure 8 indicate that after an initial transient the rate of growth decreases as the width of the initial size distribution increases. Further, Figure 10 shows that the microstructures with almost no high-angle boundaries evolved into microstructures with ~0.1 to ~0.3 fraction of high-angle boundaries depending on the width of the initial subgrain size distribution. This increase in the fraction of high-angle boundaries can be attributed to either the growth of subgrains having high-angle boundaries or subgrains with high-angle disorientations approaching each other during the course of the growth. The monotonic increase in the fraction of high-angle boundaries shows that the competition between the higher energy (Equation (3)) and higher mobility of high-angle boundaries (Equation (4)) is dominated by the role of mobility.

It was also noted that the simulations with different initial subgrain size distributions tend toward different self-similar regimes after transition from their initial state. To achieve such a condition, there must be a balance between the rate of shrinkage and then the disappearance of subgrains and the rate of growth of larger subgrains. The reason why self-similar subgrain growth occurs regardless of initial subgrain size distribution is complex and involves the interplay of boundary characteristics (energy and mobility), their boundary curvature, and local neighborhood. Figure 13 compares the evolution of the average subgrain area for different subgrain size distributions. As the width of the subgrain size distribution decreases, the growth rates increase, and self-similarity is reached faster. It is evident that the parabolic growth regime (i.e., the linear region) is obtained at shorter times as the width of the initial size distribution decreases. As a result, the total length of the high-angle boundaries decreases for the narrower distribution for longer simulation times after having reached a peak value, as shown in Figure 10a (as expected from extended grain growth).

To rationalize the observed trends for subgrain growth, the change in the average diameter of the subgrains after an increment of time from the initial state can be formulated. To perform this, one can sort the list of subgrain diameters and define a threshold index of the list for the transition from shrinkage to growth. At a given time, *t*, the average can be calculated as follows:

$$\mu_t = \frac{\sum_{i=1}^{N} d_i}{N} \tag{6}$$

where $N$ is the number of subgrains, and $d_i$ is the diameter of subgrain $i$ at time, $t$. For an incremental increase in time, the average changes as follows:

$$\mu_{t+\Delta t} = \frac{\sum_{i=1}^{KN} \left(d_i - \frac{\alpha_i}{d_i}\right) + \sum_{i=KN}^{N} \left(d_i + \frac{\alpha_i}{d_i}\right)}{N} \tag{7}$$

where $KN$ is the number of shrinking subgrains such that $K$ represent the fraction of shrinking subgrains. $\frac{\alpha_i}{d_i}$ is the shrinkage or the growth of subgrain $i$, which depends on the average curvature of the subgrain $\left(\frac{1}{d_i}\right)$ and a proportionality factor, $\alpha_i$. The $\alpha_i$ can have a different value for each subgrain depending on the subgrain average disorientation, which reflects the contribution of different subgrain boundary energies and mobilities, due to the local neighboring subgrains. Equation (7) can be simplified by using Equation (6):

$$\mu_{t+\Delta t} = \mu_t + \frac{1}{N}\left(\sum_{i=KN}^{N} \frac{\alpha_i}{d_i} - \sum_{i=1}^{KN} \frac{\alpha_i}{d_i}\right) \tag{8}$$

Here, the $\sum_{i=KN}^{N} \frac{\alpha_i}{d_i}$ term can be considered as a "gain" and the $\sum_{i=1}^{KN} \frac{\alpha_i}{d_i}$ term as a "loss" for the average. The evolution of the number fraction of shrinking subgrains ($K$ parameter) during the simulation is shown in Figure 14. Note that when $K$ equals 0.5, the population of shrinking and growing subgrains is the same.

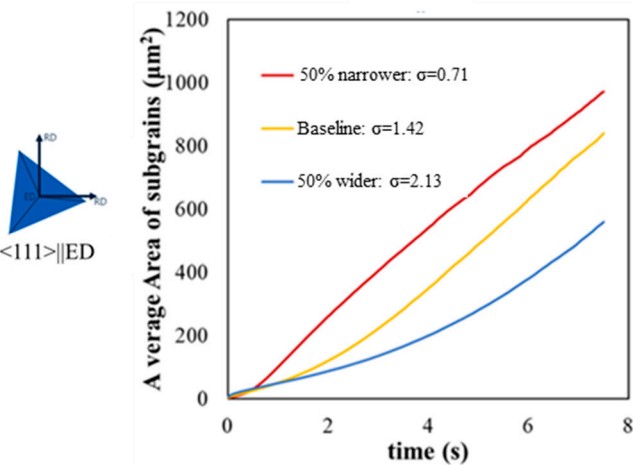

**Figure 13.** Evolution of average subgrain area as a function of time for a grain with <111>||ED orientation (note: Figure S9 in Supplementary Materials shows the results for the same analysis for <001>||ED).

An important observation from these plots is that at the end of the simulations, there is almost no change in the number fraction of shrinking subgrains, but the fraction depends on the width of the subgrain distribution, i.e., different steady-state conditions for different initial subgrain size distributions.

In all cases, $K > 0.5$, such that the shrinkage of a large population of subgrains compensates for the growth of a small population of growing subgrains, and the initial subgrain size distribution determines the transition to the self-similar regime. However, when the subgrain size distribution is initially wider, there is a larger difference between the loss and gain terms which can rationalize the initially larger rate of growth. On the other hand, for narrower distributions, the initial microstructure has a smaller difference between loss and gain terms, as the populations of shrinking subgrains and growing subgrains are closer to each other compared with the wider distribution. This can be seen by comparing the initial value of $K$ values in Figure 14, where $K$ decreases as the width of the initial subgrain size distribution decreases and is closer to 0.5 for the narrower distribution. In other words, as the initial size distribution gets narrower, the populations of shrinking and growing subgrains with similar $\frac{\alpha_i}{d_i}$ become closer to each other.

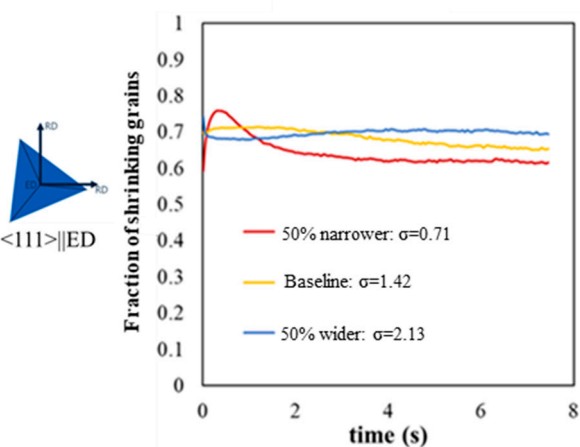

**Figure 14.** Fraction of shrinking subgrains as a function of time for <111>||ED grain. Note: Figure S10 in Supplementary Materials shows the results for the same analysis for <001>||ED).

As discussed previously for the trends in Figure 10, there is a larger population of small subgrains with large curvature for wider distribution, so there is a higher probability of having small subgrains surrounded by high-angle boundaries at the beginning of the simulation. In this light, as the initial distribution of subgrains diameter distribution gets wider, the microstructure loses more high-angle boundaries at the beginning of the simulation, which would explain the larger initial drop in the total length of high-angle boundaries shown in Figure 10a and the faster initial growth rate. However, after losing the small subgrains with high curvature, the initially wider distribution ends up with a subgrain structure with a lower average boundary curvature (i.e., a larger average diameter). Thus, there is a lower population of growing subgrains (smaller $K$) with larger $\frac{\alpha_i}{d_i}$ for wider distributions after the transition period (larger difference in gain and loss terms). This results in lower growth rates, such that the final average subgrain diameter decreases with increasing width of the initial distribution.

The emphasis of the present investigation is to rationalize key aspects of subgrain growth by analyzing the evolution of the mean subgrain size. Further details can be obtained by considering the evolution of the subgrain size distribution which is, however, beyond the scope of the present study.

*4.2. Role of Initial Subgrain Disorientation Distribution*

As shown in Figures 11 and 12, the average diameter of the subgrains grows faster as the width of the disorientation distribution increases. This can be mainly attributed to the higher probability of having subgrain boundaries with high misorientation which outgrow their surroundings and have less chance of low mobility interfaces. In fact, as the disorientation distribution gets wider, it approaches the behavior of normal isotropic grain growth for random textures [21]. A simulation was conducted with the same initial microstructure but assuming isotropic grain growth, i.e., all the boundaries have the same mobility and energy. Figure 15a compares the initial disorientation distributions for a random texture (Mackenzie distribution) and the <111>||ED grain with a halfwidth of 10.35°. The results for the evolution of the 2D subgrain/grain size are shown in Figure 15b. Subgrain/grain growth is similar in both cases, but normal grain growth results in a ~10% larger grain size. It is to be emphasized that while the disorientation distribution is wide in this case (<111>||ED), it is far from the random texture, i.e., the Mackenzie distribution. As can be read from Figure 15a., for the <111>||ED case, 88% of the boundaries have a disorientation of fewer than 15° in the initial microstructure, while it is about 2% for a random texture.

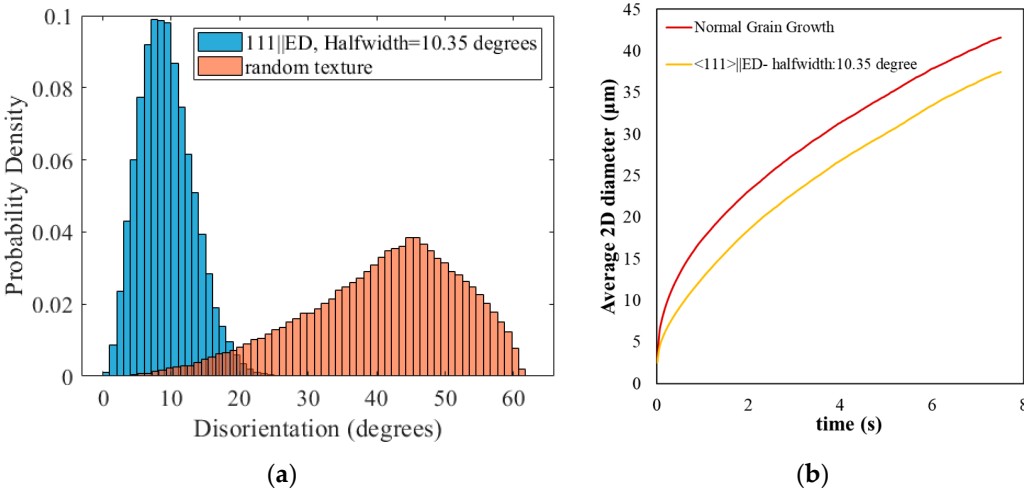

**Figure 15.** (**a**) Comparison of initial disorientation distribution in <111>||ED with halfwidth of 10.35°
with a random texture; (**b**) comparison of subgrain average diameter evolution for microstructure
with <111>||ED orientation and normal grain growth.

The effect of disorientation distribution can also be examined using the analysis
defined by Equation (8). Increasing the width of disorientation distribution changes the
threshold index of the sorted list that determines the number of subgrains that can grow, i.e.,
it increases the "gain" term. Second, it changes the proportionality factor, $\alpha_i$. As mentioned,
this factor is dependent on the average disorientation surrounding each subgrain, and it
increases as the width of the disorientation distribution increases.

It is useful to compare the distribution of effective mobility of the boundaries in the
initial microstructures with different disorientation distributions where the effective mobil-
ity is defined as $M \times \sigma$, where $M$ and $\sigma$ are the boundary mobility and energy, respectively.
Figure 16 shows the results of this analysis for different disorientation distributions. As the
initial disorientation distribution gets wider, the population of boundaries with higher effec-
tive mobility increases, and the fraction of low-mobility boundaries decreases. Specifically,
the fraction of boundaries with disorientation angles larger than 15 degrees (normalized
effective mobility of 1) is zero for the disorientation distribution with a halfwidth angle of
3.45°, and this fraction increases to 0.004 and 0.12 for halfwidth angles of 6.9° and 10.35°,
respectively. The consequence of the higher effective mobility (mobility of the boundaries
multiplied by their interfacial energy) of the boundaries is faster growth rates.

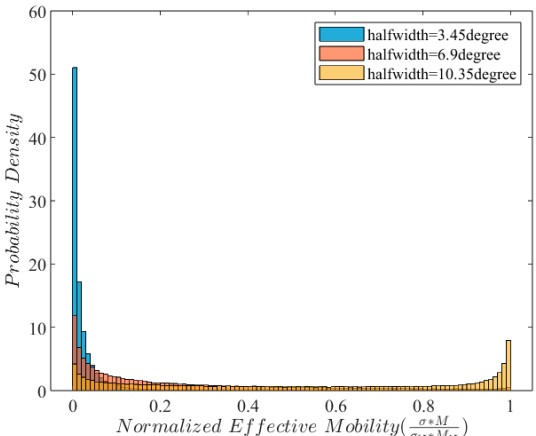

**Figure 16.** Normalized effective mobility of the initial microstructure for different initial disorientation
distributions in <111>||ED grain.

## 5. Summary and Conclusions

In this investigation, the evolution of the microstructure during subgrain growth was systematically studied using 2D multi-phase-field simulations to quantify the role of deformed state parameters, i.e., subgrain size distribution and disorientation distribution. Based on experimental observations, baseline conditions for two grains with <001>||ED and <111>||ED orientation were established, which resulted in only minor differences in the subgrain growth simulations. By systematically changing the distributions by ±50% from the baseline conditions, it was shown that both subgrain size and disorientation distributions affect subgrain growth in important ways that can be summarized as follows:

- Simulations conducted on microstructures with different subgrain size distributions (but the same disorientation distribution) show that regardless of the initial subgrain size distribution, a self-similar regime is achieved after an initial transition. However, the self-similar state is not the same for different initial size distributions. As the initial subgrain size distribution gets wider, it reaches a wider subgrain size distribution during the steady-state growth regime but with a smaller average subgrain size as compared with narrower initial size distributions.

- The effect of disorientation distribution on the evolution of subgrains is more pronounced compared with that of the subgrain size distribution. By increasing the width of the disorientation distribution, a larger increase in the average subgrain size is observed. When the cases of 50% narrower and 50% wider disorientation distributions are compared with the baseline, it is found that the average subgrain size is 8 μm smaller and 5 μm larger than the baseline, respectively, after 7.5 s of simulation time. The significant effect of the width of disorientation distribution can primarily be related to the associated mobility distributions.

In practice, it may be possible to modify the deformed state within a single grain by changing the thermomechanical history and the alloy chemistry). Further, in the case of recrystallization in materials with many grains of different preferred orientations (i.e., engineering materials such as extruded aluminum alloys), a more complex scenario needs to be considered, i.e., the interaction of elongated grains with different subgrain characteristics in terms of subgrain size and disorientation distributions, which is the subject of a separate study. Further, a specific model for the boundary characteristics has been used in the present study, and it may be useful to consider alternative models for boundary mobility and energy, e.g., benchmarked on atomistic studies. Finally, a validation of the present 2D simulations with 3D simulation would be of interest as well [55].

**Supplementary Materials:** The following supporting information can be downloaded at https://www.mdpi.com/article/10.3390/met14050584/s1, Figure S1: Different simulations to study the role of initial subgrain size distribution on the growth; Figure S2: Different simulations to study the role of initial subgrain disorientation distribution on the growth; Figure S3: Evolved subgrains structure of different microstructures with different standard deviation and the same average for <001>||ED grain; Figure S4: (a) Evolution of the equivalent area average 2D diameter of <001>||ED microstructures with different initial subgrain size distribution and (b) evolution of rate of change of the equivalent area average diameter of the same microstructures; Figure S5: Histogram of evolved normalized diameter for microstructures where <001>||ED with different initial standard deviation and same initial average diameter in different time steps; Figure S6: Evolution of (a) the total length of high angle boundaries, (b) the fraction length of the high angle boundaries of single grains in <001>||ED; Figure S7: The evolved microstructure of a grain in <001>||ED fibre with the same subgrain size as baseline and disorientation distribution of different half-width angles (a) 4.5°, (b) 9° (baseline), (c) 13.5° at 7.5 s and (d) evolution of diameters for different half-width angles; Figure S8: The evolution of average disorientation in the different microstructure of different disorientation distribution with the same subgrain size in a grain with <001>||ED; Figure S9: Evolution of subgrains area as a function of time for a grain with <001>||ED; Figure S10: Fraction of shrinking grains as a function of time for <001>||ED grain; Table S1: Key parameters for the initial microstructures synthesized to study the role of subgrain size distribution; Table S2: Half-width

angle in degree for ODF calculation of the initial microstructures synthesized to study the role of disorientation distribution.

**Author Contributions:** Conceptualization, A.K., W.J.P., M.G. and M.M.; data curation, A.K.; formal analysis, A.K.; funding acquisition, W.J.P.; investigation, A.K.; methodology, A.K., W.J.P., M.G. and M.M.; project administration, W.J.P.; resources, W.J.P., M.G. and M.M.; software, A.K. and M.G.; supervision, W.J.P., M.G. and M.M.; validation, A.K., W.J.P., M.G. and M.M.; visualization, A.K.; writing—original draft, A.K.; writing—review and editing, A.K., W.J.P., M.G. and M.M. All authors have read and agreed to the published version of the manuscript.

**Funding:** This research was funded, in part, by the Natural Sciences and Engineering Research Council (NSERC):NSERC-RGPIN-2019-04043.

**Data Availability Statement:** The raw data supporting the conclusions of this article will be made available by the authors on request.

**Acknowledgments:** This work was undertaken, in part, thanks to funding from the Canada Research Chair program (Poole). Support of Rio Tinto Aluminum and the Natural Sciences and Engineering Research Council of Canada (NSERC) is gratefully acknowledged. This research was also enabled in part by support provided by the Digital Research Alliance of Canada ( alliancecan.ca) and all regional partner organizations (ACENET, Calcul Québec, Compute Ontario, the BC DRI Group and Prairies DRI).

**Conflicts of Interest:** The authors declare no conflicts of interest.

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
