# Peer review of "Large-Scale Multi-Phase-Field Simulation of 2D Subgrain Growth"

_metals, doi:10.3390/met14050584_

Round 1

Reviewer 1 Report

Comments and Suggestions for Authors

The manuscript is a computationally study of the sub-grain growth behaviour of the sub-grain structure in an individual grain representing a single crystallographic grain resulting from extrusion of an Al 3xxx alloys (i.e. hot deformation).

Large-scale 2D phase field simulations have been conducted by systematically varying the initial subgrain size and disorientation distributions, from that of a base line structure (obtained from experiment).

The paper is well structured and well written, and the results are presented in mainly nice informative figures and graphs. And given the scope of the paper the results are carefully and thoroughly analyzed and discussed and the conclusions safe and sound.

However, although an interesting computational exercise that provides some intriguing results, it is somewhat artificial and detached from reality, which rises some questions:

What can we learn from this computationally study, and in what way or to which extent do the results have any possible implications in the control or tuning of processing conditions during and after extrusion and possible accompanying effects on microstructure and properties?

Is it possible to manipulate the sub-structure evolution during extrusion to somehow vary influence the sub-grain and misorientation distribution?

Are any sub-structure characteristics more beneficial than others?

A related more technical question: Is it possible to have sub-grain structures have the same disorientation distribution, but still being different (sub-grains which locally have different neighborhoods)?

As also indicated by the authors, for further work it would be interesting to also investigate different scenarios wrt grain boundary energy and mobility, including the characteristics of special boundaries (Sigma-boundaries), and also to do 3D simulations?       

Some more specific questions/comments to the text:

p.2, l 75 …; Recrystallization in aluminum alloys deformed at high temperatures is a continuous process that results as a consequence of subgrain structure coarsening to reduce the stored energy in the material [26–29] and this constitutes anisotropic (sub)grain growth.

Do the authors by this statement indicate that ‘recrystallization’ after hot deformation (like extrusion) takes place as an extended recovery process.

Considering Fig. 12, the resulting (sub-)grain structure after 7.5 s (in terms of disorientation distribution) is still very far from a structure dominated by high angel boundaries!?

p. 3, line 138 …

All the histograms of disorientation distributions in this study are the pixel weighted disorientation distribution which can have a value up to 30o due to the chosen 15o threshold angle for grain identification.

This statement is not quite clear to present reviewer ?

 Re Fig. 2:

What are the meaning of the different colors in Fig. 2b, i.e. what is the difference between the 4 quadrants?

Re Fig. 11 and 12.

Like in several other figures, it could be useful to name the variant with halfwidth = 6.9, the base line case, to clearly indicate the reference referred to in the text!   

Reviewer 2 Report

Comments and Suggestions for Authors

Comments to the manuscript “Large scale multiphase field simulation of 2D subgrain growth” by Ali Khajezade et al. The manuscript employed the phase-field method to simulate the subgrain growth of aluminum alloys after deformation in a quite large scale with amounts of grains. Apparently, recrystallization behavior of grains after deformation depends strongly on the stored energy, here in this work, the authors assumed that the stored energy and the corresponding dislocations are relaxed to the subgrain structures. Therefore, the simulations only considered the initial subgrain size and the disorientation on the grain growth during recrystallization. This simplification leads to the coarsening of subgrains in simulations driving only by curvature and grain boundary energy. Is this simplification reasonable for aluminum alloys? This is the main point I want to raise. However, this simplification makes the phase-field model and the simulation much more easily to be performed to study the recrystallization. At least the simulations showed the grain growth behavior with different initial grain size and disorientation distribution. So I recommend minor revision of the manuscript before publication. Some comments may be considered in the revisions.

(1)  The initial grain size is about 2μm, and the diffuse interface width is chosen as 6cells*0.25μm=1.5μm. The interface width is too large compared to the grain size. This too large width will lead to different grain growth behavior. I suggested the authors should make convergence test of the interface width parameters. For example initializing 100 subgrains with different interface width parameters and to see whether the average grain size converged when reaching steady-state coarsening. So that a converged interface parameters can be obtained and makes the simulations being more soundly.

(2)  Please give more detail of the algorithm to calculate the millions of grains using multi-phase field model. To apply each grain with an order parameter is impossible. In that case you should solve millions of eq.(2) for millions of grains and the computing efficiency is very very low.

(3)  Making the results and discussion more concentrated to the topic, presently, it seems verbose.

Reviewer 3 Report

Comments and Suggestions for Authors

 The manuscript clarified the effect of the initial subgrain size and disorientation distribution on the growth by large scale multiphase field simulation. The size and disorientation distribution were measured and modelled in multiphase field simulation. The disorientation dependence of the grain boundary energy and mobility was implemented to the model. To investigate the effect of them, they intentionally change the size and disorientation distribution while the other microstructural characteristics including average grain size remain unchanged. As the result, the author found that 1. self-similar regime is achieved in any size distribution. 2. As the initial size distribution is wider, wider steady state is achieved with smaller average size. 3. The effect of disorientation distribution is more pronounced compared to that of size distribution. Most of the methodology is appropriate and the conclusions derived from these results and the interpretations of the calculation results are consistent and sound. The paper would be of interest to the readers of Metals but I have one major comment as:

 1. Line 207, the authors set the interface width and mesh size as 1.5 and 0.25 μm, respectively, while average subgrain size is 2.06 μm. According to the Fig. 6a, most grains are less than interface width and some grains are one or two meshes in the cases of σ= 1.420 and 2.130. Can this calculation correctly capture the subgrain growth including curvature effect?

 I think this paper would be acceptable for publication if the above point is addressed convincingly. The minor comments and questions are listed below:

2. Line 350 to 353, Figure 12, instead of 10, should be referred or one figure similar to Fig. 12 is missing.

3. Line 370 to 372, can the fraction of high-angle boundaries be extracted?

4. Line 460 to 472, can the temporal evolution of the subgrain size distribution be extracted?

5. Figure 16, can the distribution explain only 10% difference in subgrain/grain size with normal grain growth? Temporal evolution of the effective mobility distribution might clarify the mechanism.

Reviewer 4 Report

Comments and Suggestions for Authors

This manuscript is focused in the analysis of Al alloys 2D subgrain growth. The authors apply large scale multiphase field simulation.

There are all references in the methods applied. For example, the methods applied for fitting with an orientation distribution functions (ODF). New methods and approaches developed in the last years should be applied (it can be compared with these all methods).

The authors remark the interest of Al alloys as Al-Mn-Fe-Si. In the quaternary systems the role of each element is very important. In this work the effect of the different elements is not shown in the simulation. Thu, I assume that is an ideal simulation. The authors should justify the interest.

I also recommend improving the discussion by adding modern references.

Minor comment: Figure 1 is not necessary.

Thus, I recommend the rejection.

Comments on the Quality of English Language

Minor editing of English language required

Round 2

Reviewer 3 Report

Comments and Suggestions for Authors

The manuscript has been revised well. I think this manuscript is acceptable now.

Reviewer 4 Report

Comments and Suggestions for Authors

I recommend to accept this manuscript

Comments on the Quality of English Language

Minor editing of English language required